# The Relationship between the Frequency of Breakfast Consumption, Conversation with Parents, and Somatic Symptoms in Children: A Three-Wave Latent Growth Model

**DOI:** 10.3390/ijerph191912975

**Published:** 2022-10-10

**Authors:** Shin-Il Lim, Sookyung Jeong

**Affiliations:** 1Department of Educational Psychology, College of Nursing, Jesus University, Jeonju 54989, Korea; 2Department of Nursing, College of Medicine, Wonkwang University, Iksan 54538, Korea

**Keywords:** breakfast consumption, somatic symptoms, children’s health, parent–child conversation, longitudinal study

## Abstract

Breakfast consumption is essential for children to generate energy for the day. Parents play an important role in children’s breakfast habits and spending time with parents during breakfast greatly influences children’s emotional development. Therefore, this study investigated the relationship between the frequency of children’s breakfast consumption, time spent in conversation with parents, and children’s somatic symptoms. Data were obtained from the Korea Children and Youth Panel Survey 2018 and were initially collected from fourth-grade elementary school students and followed up for three years. SPSS 21.0 and AMOS 21.0 software were used for data analysis. Multivariate latent growth modeling was applied to analyze the effect of the breakfast consumption frequency on children’s somatic symptoms and the mediating effect of parent–child conversation time on this relationship. Consequently, as children’s frequency of breakfast consumption increased, their somatic symptoms decreased. Furthermore, parent–child conversation time mediated the relationship between these two variables. Therefore, students, parents, and teachers should be educated about the importance of having breakfast and communicating with parents for students’ emotional health.

## 1. Introduction

Children need to consume a balanced diet as they require abundant energy and nutrients to perform vigorous physical activities and perform academic activities. Hence, regular breakfast consumption is essential for the same. Breakfast consumption has been a major component of children’s diet as it supplies nutrients and energy necessary for the initial stage of body metabolism after an empty fasting time [1]. Research has confirmed that breakfast consumption influences children’s mental health, academic performance, and risk of obesity [2]. Additionally, a survey of Spanish adolescents shows that breakfast quality can influence mental health, including stress and depression [3]. Moreover, the prevalence of children’s breakfast consumption is influenced by their parents’ perceptions, breakfast eating habits [4], and family structure, including living with both parents or a single parent [5]. Thus, parents play an essential role in children’s breakfast consumption.

A survey of Korean children reveals that only 53.7% of Korean parents talk to their children daily. Furthermore, on average, Korean children spend only 13 min per day talking with family members [6]. The more time children spend talking to their parents, the more positively they perceive them. Parents’ positive parenting attitudes affect children’s depression, social withdrawal, and life satisfaction. Therefore, it is crucial to secure conversation time between parents and children [7,8]. In Korea, 47.5% of conversations with children occur during meals [6]. A Korean survey revealed that 11.9% of children never had a meal with their family for a week. Furthermore, children who ate with their family one to four times a week accounted for 34.6%, while those having five to six times a week or more than once a week were 12.2% and 41.3%, respectively [9]. It was surveyed that Koreans often eat dinner with their families than breakfast or lunch [10].

Somatization refers to the tendency to experience physical discomfort and pain in the absence of a clear physiological cause and to seek medical help in responding to psychological stress [11]. Previous studies have verified that psychological and emotional factors, including stress and depression, are major predictors of somatic symptoms [12,13]. Somatic symptoms may develop because people have a low ability to consciously experience and appropriately express negative emotions, such as stress, depression, and anger [14]. Compared with adults, it is difficult for children and adolescents to express their emotions through language, making them vulnerable to somatic symptoms [15]. Parents and children share one of the most complex biological, psychological, and social relationships, and parents greatly influence children’s emotionality, especially as they help them learn emotion regulation [16].

Parents have the ability to exert a positive influence on children’s emotional problems, which can eventually prevent their somatic symptoms. Therefore, it is important to secure frequent parent–child interactions in daily life. Studies have examined the role of parenting stress in the relationship between children’s breakfast habits and physical and emotional development [17,18]. However, to our knowledge, no study has examined the relationship between the frequency of breakfast consumption, conversation time with parents, and somatic symptoms in children. Additionally, childhood is an unstable period where physical and emotional changes occur rapidly, through a wide variety of influences [19]. Therefore, using a longitudinal study design, we examined how the frequency of breakfast consumption, conversation time with parents, and somatic symptoms change, and how these factors influenced each other from the 4th to the 6th grade of elementary school.

### Research Questions

Figure 1 shows the research model for this study. This model evaluated the initial value, rates of change in each variable (inclination), and the relationship between all the variables as per the inclination. This study examined the mediating effect of conversation time with parents on the relationship between children’s frequency of breakfast consumption and somatic symptoms over three years.

Research Question 1. What were the changes in the children’s frequency of breakfast consumption, somatic symptoms, and conversation time with parents over the three years?

Research Question 2. What effect did the children’s frequency of breakfast consumption have on their conversation time with parents and children’s somatic symptoms over the three years?

Research Question 3. How did conversation time with parents affect somatic symptoms in children over the three years?

Research Question 4. Did conversation time with parents mediate the relationship between the frequency of breakfast consumption and somatic symptoms in children over the three years?

## 2. Materials and Methods

### 2.1. Study Participants

This study used longitudinal panel data from the first (fourth-grade elementary school students), second (fifth-grade elementary school students), and third years (sixth-grade elementary school students) of the Korea Children and Youth Panel Survey 2018 (KYCPS 2018) [20], conducted by the National Youth Policy Institute. A multi-stage stratified cluster sampling method was used to construct the original sample following four steps. Step 1: based on a population analysis, 17 cities and provinces in Korea, 6040 schools nationwide, and 471,566 students were identified as the total participants. Step 2: stratified sampling was used as it can provide population representativeness despite a small sample size. Step 3: sample allocation was proportionally distributed based on the number of students in the 17 provinces, assuming 70% effective responses on average. Step 4: the schools were selected using probability proportional to size sampling. This study collected data via individual interviews (TAPI—Telephony application program interface) using tablets and personal computers. Following the data utilization standards of the National Youth Policy Institute, the institution’s name and title of the investigation were specified, and personal information related to the participants was excluded. Table 1 describes the general characteristics of the participants.

### 2.2. Measurements

#### 2.2.1. Frequency of Breakfast Consumption

We selected one question on breakfast consumption from the KCYPS 2018: “How many times did you eat breakfast last week?”

#### 2.2.2. Parent–Child Conversation

Questions about parent–child conversations in the KCYPS 2018 were selected and revised. These included two questions assessing “talk time with parents” on weekdays and weekends. A high score revealed more time spent talking with parents. The combined values of weekdays and weekends were used. Example: “How much time do you spend talking with your parents during the week/weekend?”

#### 2.2.3. Somatic Symptoms

Somatic symptoms were evaluated using items from the KCYPS 2010, adapted from Cho and Lim (2003) [21]. These questions were revised and supplemented. Eight questions were rated from 1 (not at all) to 4 (strongly agree). A high score revealed more somatic symptoms than a low score. Example: “I often have a headache.” The original scale’s reliability was at a Cronbach’s alpha of 0.810, while in this study, the reliability was 0.842, 0.861, and 0.892 in the first, second, and third years of the KCYPS, respectively.

### 2.3. Data Analysis

Multivariate latent growth modeling was applied to analyze the effect of the frequency of breakfast consumption on children’s somatic symptoms, mediated by parent–child conversation time. Furthermore, we verified the research questions and constructed the research model (Figure 1). The latent growth model uses longitudinal data to confirm the changes of each variable over time, and estimates the averages of the intercept and slope of each variable and the path coefficient between variables through model analysis. In the latent growth model, the intercept refers to the average value at the time of observation, and the slope of each variable refers to the degree of change in the average over time.

Through the analysis of the latent growth model, we can estimate the changes in each variable according to the growth of elementary school students and the relationship between these changes. First, we define a function to explain changes in variables simply and examine whether changes in individual differences are significant. Change is a linear intercept and a function of slope. In this case, each intercept’s individual difference refers to the individual difference in the initial status of each variable. Furthermore, the slope of the individual difference refers to the individual difference in the change rate of each variable [22].

This study verified the research question by setting the intercept (initial value) and slope (change rate) of elementary school students’ breakfast frequency, parental conversation time, and somatic symptoms as latent variables. The measured variables of each latent variable were the three variables measured in the third year (2020).

Data were analyzed using SPSS Version 21.0 (IBM Corp., New York, NY, USA) and AMOS 21.0 software (IBM Corp., New York, NY, USA). The latent growth model fit was verified through chi-square (χ^2^), Comparative Fit Index (CFI), Tucker–Lewis Index (TLI), and Root Mean Square Error of Approximation (RMSEA) analyses. As χ^2^ is sensitive to the sample size, this study focused on CFI, TLI, and RMSEA. CFI is not sensitive to the fit index and sample size, and TLI is characterized by considering the model’s simplicity. Therefore, the model fit can be judged by considering CFI and TLI simultaneously [23]. In RMSEA, a value less than 0.05 is considered a very good fit, less than 0.80 is a good fit, and less than 0.10 is an average fit. Conversely, CFI and TLI values greater than 0.90 are good fit indices [24]. Hence, the interpretation was performed accordingly. Due to the longitudinal nature of the study, there was a possibility of selection bias caused by missing values. Therefore, we resolved the selection bias by using the full information maximum likelihood method (FIML) to analyze the model.

## 3. Results

### 3.1. Descriptive Statistics for the Study Variables

Table 2 shows the results of descriptive statistics of this study. The frequency of breakfast consumption in elementary school students was over five times a week, but it gradually decreased with advancing grades. The amount of parental-child conversation time was, on average, four hours a week, gradually decreasing as the grade advanced. Somatic symptoms were the lowest in the fourth grade, gradually increasing thereafter. Table 2 presents the mean, standard deviation, minimum and maximum values, skewness, and kurtosis of the variables analyzed in this study. Descriptive statistics indicated that the means and standard deviations ranged from 1.6912 to 5.8550 and 0.5845 to 2.0489, respectively. The minimum and maximum values of all variables were 0–1 and 3.88–7.0, respectively. Additionally, for psychometric purposes, skewness and kurtosis values between −2 to +2 are acceptable [25,26]. Comprehensively, the assumption of normality was satisfied.

### 3.2. Correlations and Multicollinearity among the Variables

Table 3 presents the correlations and multicollinearity among the major variables. The correlation between the frequency of breakfast consumption and parent–child conversation was r = 0.1046 (*p* < 0.001) in the fourth grade, r = 0.00699 (*p* < 0.01) in the fifth grade, and r = 0.1546 (*p* < 0.001) in the sixth grade, showing a positive correlation with a V-shaped curve. The correlation between the frequency of breakfast consumption and children’s somatic symptoms was r = −0.1596 (*p* < 0.001) in the fourth grade, r = −0.1665 (*p* < 0.001) in the fifth grade, and r = −0.2058 (*p* < 0.001) in the sixth grade, revealing a negative correlation between the two variables and an upward-sloping shape. The correlation between parent–child conversation time and children’s somatic symptoms was r = −0.1644 (*p* < 0.001) in the fourth, r = −0.1078 (*p* < 0.001) in the fifth grade, and r = −0.2763 (*p* < 0.001) in the sixth grade. The two variables had a negative correlation and the sixth graders showed the highest V-shaped curve. The variance expansion index (VIF) ranged from 1.215 to 1.4893, with all values below 10. The tolerance was higher than 0.1, satisfying multicollinearity.

### 3.3. Analyzing the Study Model

Statistical analyses showed that the mean of the children’s frequency of breakfast consumption, parent–child conversation time, and children’s somatic symptoms at the three time points (fourth, fifth, and sixth grade) changed consistently over time. We applied the no-change and linear change models to each factor to estimate the changes in the three factors for evaluating the development trajectory of each variable. The no-change model assumes that the mean does not change over time; only the initial value is given in this model. The linear model is applied when the mean change of a factor constantly increases or decreases with time. Table 4 shows the fitness results that determined whether or not there was a model change for each variable.

As for the fitness of the no-change model and the linear change model for each variable presented in Table 4, the fitness of the linear change model for all variables was relatively satisfied. Furthermore, considering the parameter estimates (the breakfast consumption frequency, parent–child conversation, and somatic symptoms), the initial mean frequency of breakfast consumption and the mean rate of change showed significant differences: 5.8463 (*p* < 0.001) and −0.3022 (*p* < 0.001), respectively. This means that the breakfast consumption frequency gradually decreased over time from the fourth to the sixth grade of elementary school. The covariance between the initial value of the breakfast consumption frequency and the mean rate of change was −0.4342 (*p* < 0.001), while the correlation coefficient was −0.4123. The mean rate of change and the covariance values were negative. Therefore, the rate of change in the breakfast consumption frequency decreases slowly when the initial value of the breakfast consumption frequency is low. Conversely, when the initial value is high, the rate of change decreases rapidly.

The mean initial value of parent–child conversation time was 4.2872 (*p* < 0.001), and the mean rate of change was −0.0261 (*p* < 0.05), indicating a significant difference. This implies that conversation time between parents and children decreased from the fourth to the sixth grade of elementary school. The covariance between the initial value of parent–child conversation time and the mean rate of change was −0.1897 (*p* < 0.001) and the correlation was −0.4295. The mean rate of change and the covariance values are negative. Therefore, the rate of change in parent–child conversation time decreases slowly when the initial value of parent–child conversation time is low. However, when the initial value is high, the rate of change decreases rapidly.

The mean initial value of children’s somatic symptoms was 1.6935 (*p* < 0.001), and the mean rate of change was 0.1492 (*p* < 0.001). This means that somatic symptoms gradually increased from the fourth to the sixth grade of elementary school. The covariance between the initial value and the mean rate of change was −0.0142 (*p* < 0.05) and the correlation was −0.3401. The mean rate of change is positive (+) and the covariance is negative (−). Therefore, the rate of change increases rapidly when the initial value of somatic symptoms is low. However, when the initial value is high, the rate of change increases slowly. Finally, the initial value and covariance of the change rate of the breakfast consumption frequency, parent–child conversation time, and somatic symptoms showed significant differences. This confirmed that individual differences existed in the initial value and rate of change in the development trajectories of all three variables.

### 3.4. Verification of the Research Model

In the final *multivariate latent growth model*, the relationship of each variable was established based on the results of the analysis model of the variables (Figure 2). Furthermore, Table 5 reveals that the fitness of the final model was satisfactory.

Table 6 illustrates the path coefficients for each route, which showed how the breakfast consumption frequency affected parent–child conversation time and children’s somatic symptoms and how the parent–child conversation time affected somatic symptoms. The initial value of the breakfast consumption frequency was found to have a significant positive effect on the initial value of parent–child conversation time (β = 0.1723, *p* < 0.001). This implies that a high frequency of breakfast consumption resulted in more time spent talking with parents. The initial value of the frequency of breakfast consumption did not significantly affect the rate of change in parent–child conversation time. However, the rate of change in the breakfast consumption frequency had a significant positive effect (β = 0.3227, *p* < 0.001) on the rate of change in parent–child conversation time. Thus, a comprehensive examination of the relationship between the initial value and the rate of change, reveals a close positive relationship between the breakfast consumption frequency and the amount of parental-child conversation time during the development of elementary school students.

The initial value of the frequency of breakfast consumption was found to have a significant negative effect on the initial value of children’s somatic symptoms (β = −0.2332, *p* < 0.001). This means that a high breakfast consumption frequency results in few somatic symptoms. The initial value of the frequency of breakfast consumption did not significantly affect the rate of change in somatic symptoms. However, the rate of change in the breakfast consumption frequency had a significant negative effect (β = −0.2174, *p* < 0.001) on the rate of change of somatic symptoms. Therefore, a comprehensive examination of the relationship between the initial value and the rate of change reveals a strong negative relationship between the breakfast consumption frequency and somatic.

The initial value of parent–child conversation time was found to have a significant negative effect on the initial value of somatic symptoms (β = −0.2524, *p* < 0.001). This implies that the more time parents spend talking with their children, the fewer the children’s somatic symptoms. The initial value of parent–child conversation time did not significantly affect the rate of change of somatic symptoms. However, the rate of change in parent–child conversation time had a significant negative effect (β = −0.6262, *p* < 0.001) on the rate of change of somatic symptoms. This means that the greater the decrease rate of parent–child conversation, the greater the rate of increase in children’s somatic symptoms. A comprehensive examination of the relationship between the initial value and the rate of change revealed a very strong negative relationship between the amount of parental-child conversation time and somatic symptoms during the development of elementary school students.

Table 7 shows the longitudinal mediating effect between the variables through the Sobel test based on the results shown in Table 6. The frequency of breakfast consumption had a significant negative effect on somatic symptoms mediated by parent–child conversation time. Furthermore, the rate of change in the frequency of breakfast consumption had a significant effect on the rate of change in somatic symptoms mediated by the rate of change in parental-child conversation time.

## 4. Discussion

This study evaluated a latent growth model that verified the relationship between children’s breakfast consumption frequency, parent–child conversation time, and somatic symptoms in children over three years.

The results show that the frequency of children’s breakfast consumption decreased as they grew old. This is consistent with the Health Behavior in School-aged Children Survey in a Scottish study [27]: as children entered higher school grades, irregular breakfast consumption increased by a maximum of 3.13 times. Irregular breakfast consumption is closely related to insufficient sleep and increased screen time [28]. Studies show that students in higher grades reported spending more time playing games and watching TV [29] and less sleep time [30]. Additionally, Vereecken et al. (2009) reported regular breakfast consumption in school-aged children to be positively associated with a healthy lifestyle, including increased physical activity, watching TV for less than two hours a day, and daily consumption of fruits and vegetables, across 41 countries [31]. In other words, irregular breakfast consumption is closely related to an unhealthy lifestyle. Moreover, irregular breakfast consumption causes low energy levels in the morning and low participation in physical activities [32,33] and may lead to obesity in children [34]. Therefore, school nurses, teachers, and parents should mandatorily educate children about the importance of a healthy lifestyle and regular breakfast consumption at an early age.

Somatic symptoms occur when an individual’s subjective symptoms (causing functional impairment) are not consistent with a physical disease or etiology [9]. These symptoms may occur in various forms, abdominal pain and headache being the most common in children [35]. The present study showed that children’s breakfast consumption frequency influenced their somatic symptoms. As the breakfast consumption frequency in a week increased, somatic symptoms significantly decreased. Similarly, Azemati et al.’s (2020) study demonstrated that an increased number of skipped breakfasts were associated with a high number of somatic symptoms and psychological health complaints in Iranian children and adolescents [36]. Contrarily, a study on Japanese school children reported that skipping breakfast had no significant correlation with somatic symptoms [37]. Despite the contradictory findings of some studies, somatic symptoms in children are vital issues to be addressed. According to Nakao (2006), it is crucial to determine the factors influencing somatic symptoms and control them as they are a risk factor for depression [38].

This study indicated that the number of breakfasts consumed by children was positively related to time spent in conversing with their parents, decreasing the children’s somatic symptoms. That is, as the children’s breakfast consumption frequency increased, the amount of conversation time with their parents increased. Additionally, conversation time with parents influenced children’s somatic symptoms. So far, studies have barely evaluated the mediating effect of parent–child conversations on the relationship between children’s breakfast consumption and their somatic symptoms. However, Bae’s study (2013) reported a positive relationship between the frequency of eating a family meal and communication with parents [39]. It is well acknowledged that children are vulnerable to discerning physical pain and emotional pain [40]. Furthermore, children tend to complain of physical pain instead of emotional pain if self-expression is suppressed in their homes [41]. Therefore, parents must identify their children’s problem behaviors (including irritability, crankiness, and apathy) caused due to emotional problems. These children’s physical languages are closely related to somatic symptoms [41]. Eating breakfast together helps create love, warmth, and empathy in families [42] and provides opportunities to identify their behavioral problems associated with somatic symptoms. Family members must spend time together in their daily lives for an ideal and healthy family relationship. Eating breakfast together allows one to communicate with each other and enhances family intimacy. Consequently, strong relationships developed among family members during family breakfasts can improve children’s mental health and decrease depression [43,44]. Therefore, when developing a model to improve children’s mental health, emphasis should be on time spent with parents and strong family relationships through communication.

This study has several limitations. First, the data analyzed were obtained from a longitudinal survey that targeted fourth grade elementary school students during initial data collection. Therefore, the findings may not apply to all elementary school students. Second, this study did not deal with the content and quality of conversations between parents and children. Therefore, future research should evaluate these aspects to extend the findings.

## 5. Conclusions

This longitudinal study analyzed survey data to reveal the relationships between children’s breakfast consumption frequency, conversation time with parents, and children’s somatic symptoms over three years. It was found that children’s somatic symptoms decreased as their breakfast consumption frequency increased, and parent–child conversation time mediated the relationship between these two variables. However, as children progressed to higher school grades, the breakfast consumption frequency and conversation time with parents decreased, and somatic symptoms increased. This study contributed to understanding the close relationship between breakfast consumption, conversation time with parents, and children’s emotional problems. Starting the day by having breakfast with one’s family is meaningful because it simultaneously helps satisfy the psychological and nutritional needs of family members.

Various programs aim to educate parents about children’s emotional problems, yet the problems continue to increase. Parents probably find education-oriented programs difficult to follow. Moreover, the programs’ contents may not be suitable for each family. Instead, task-oriented interventions for parents may prove a feasible alternative for the same. Based on the current study, eating breakfast together can be a possible solution to children’s emotional problems. Parents should spend adequate time with their children in the morning for children’s emotional health and to build cohesive relationships. Additionally, children with increasing somatic symptoms showed a greater decrease in the breakfast consumption frequency. Moreover, the greater the decrease in parent–child conversation time, the greater the increase in somatic symptoms. Therefore, schools should educate students, parents, and teachers about the importance of having breakfast and adequate conversation time with parents. Special attention should be paid to these factors in the case of children with somatic symptoms.

## Figures and Tables

**Figure 1 ijerph-19-12975-f001:**
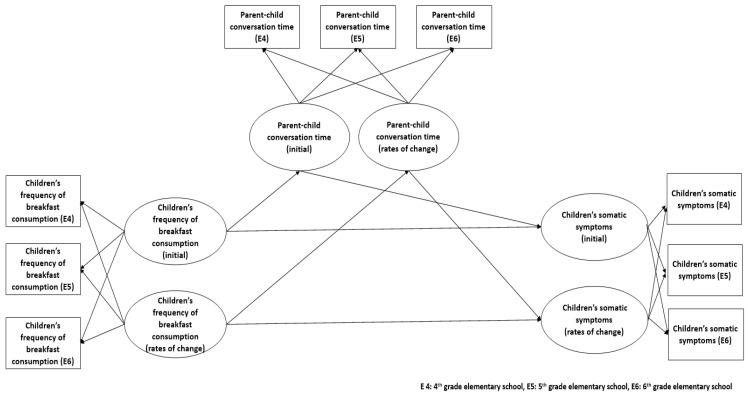
Research model.

**Figure 2 ijerph-19-12975-f002:**
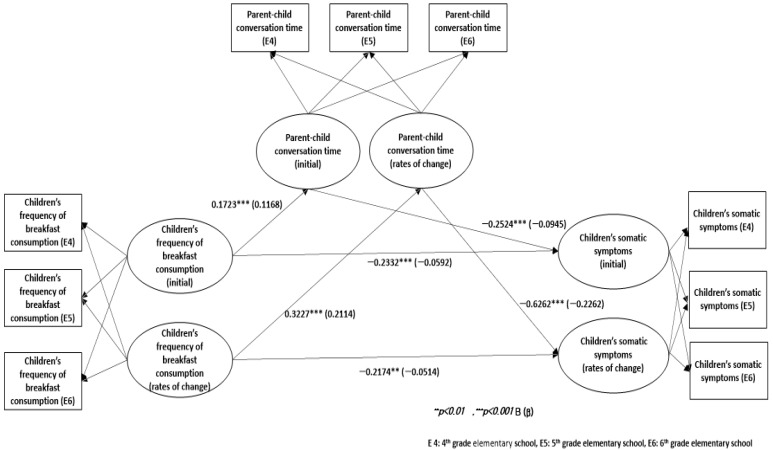
The results of latent growth analysis in the frequency of breakfast consumption, the amount of parental-child conversation time and somatic symptoms.

**Table 1 ijerph-19-12975-t001:** General characteristics of participants.

	1st Year (2018) 4th Grade/11 Years Old	2nd Year (2019) 5th Grade/12 Years Old		3rd Year (2020) 6th Grade/13 Years Old	
	The Number of Participants (A)	Remaining Number of Participants	Maintaining Rate about Original Sample	Remaining Number of Participants	Maintaining Rate about Original Sample
total	2607	2437	93.5	2411	92.5
male	1313	1221	92.9	1211	92.2
female	1294	1216	93.9	1200	92.7

**Table 2 ijerph-19-12975-t002:** Statistics for the major variables.

	n	Mean	SD	Skewness	Kurtosis
CFBC (4th grade/11 years old)	2607	5.8550	1.9374	−1.6530	1.6239
CFBC (5th grade/12 years old)	2437	5.5224	2.0159	−1.2725	0.5905
CFBC (6th grade/13 years old)	2411	5.2559	2.0489	−1.0094	0.0467
APC (4th grade/11 years old)	2607	4.2919	1.5639	0.1390	−0.9529
APC (5th grade/12 years old)	2437	4.2682	1.4218	0.2362	−0.7395
APC (6th grade/13 years old)	2411	4.2561	1.4725	0.2844	−0.8074
CSS (4th grade/11 years old)	2607	1.6912	0.5774	0.6720	−0.14555
CSS (5th grade/12 years old)	2437	1.7611	0.5845	0.6280	−0.0565
CSS (6th grade/13 years old)	2411	1.8250	0.6390	0.5403	−0.4423

CFBC: Children’s frequency of breakfast consumption, APC: Amount of parent–child conversation time, CSS: Children’s somatic symptoms.

**Table 3 ijerph-19-12975-t003:** Relationships between constants and multicollinearity statistics.

		②	③	④	⑤	⑥	⑦	⑧	⑨
CFBC (4th grade)	1.0000								
② CFBC (5th grade)	0.4580 ***	1.0000							
③ CFBC (6th grade)	0.3499 ***	0.4653 ***	1.0000						
④ APC (4th grade)	0.1046 ***	0.0791 ***	0.0118	1.0000					
⑤ APC (5th grade)	0.0585 **	0.0699 **	0.0582 **	0.3659 ***	1.0000				
⑥ APC (6th grade)	0.0857 ***	0.1080 ***	0.1546 ***	0.2844 ***	0.4038 ***	1.0000			
⑦ CSS (4th grade)	−0.1596 ***	−0.0916 ***	−0.0648 **	−0.1644 ***	−0.0633 **	−0.0906 ***	1.0000		
⑧ CSS (5th grade)	−00763 ***	−0.1665 ***	−0.0972 ***	−0.0679 **	−0.1078 ***	−0.1037 ***	0.3608 ***	1.0000	
⑨ CSS (6th grade)	−0.0920 ***	−0.1030 ***	−0.2058 ***	−0.0490 *	−0.0934 ***	−0.2763 ***	0.2548 ***	0.3780 **	1.0000
tolerance	0.7467	0.6714	0.7258	0.8213	0.7680	0.7520	0.8227	0.7623	0.7676
VIF	1.3392	1.4893	1.3777	1.2176	1.3021	1.3298	1.2155	1.3117	1.3027

CFBC: Children’s frequency of breakfast consumption, APC: Amount of parent–child conversation time, CSS: Children’s somatic symptoms; *** *p* < 0.001, ** *p* < 0.01, * *p* < 0.05.

**Table 4 ijerph-19-12975-t004:** Model fitness for each variable and the results of the models.

Model		*x* ^2^	*df*	*p*	*TLI*	*CFI*	*RMSEA*	Mean		Variance		Covariance/Correlation
								Initial	Rate of Change	Initial	Rate of Change	
CFBC	no-change	30.6102	4	0.000	0.7283	0.8192	0.1454	5.5664 ***		1.6801 ***		
	linear change	1.0277	1	0.311	0.9701	0.9713	0.0795	5.8463 ***	−0.3022 ***	2.1654 ***	0.4527 ***	−0.4342 ***/−0.4123
APC	no-change	49.9408	4	0.004	0.9791	0.9861	0.0332	4.2702 ***		0.7859 ***		
	linear change	0.2135	1	0.213	1.0063	1.0000	0.0000	4.2872 ***	−0.0261 *	0.9599 ***	0.1663 ***	−0.1897 ***/−0.4295
CSS	no-change	113.6001	4	0.000	0.7782	0.8521	0.1031	1.7518 ***		0.1193 ***		
	linear change	11.1895	1	0.665	1.0004	1.0079	0.0000	1.6935 ***	0.0665 ***	0.1492 ***	0.0376 ***	−0.0142 */−0.3401

CFBC: Children’s frequency of breakfast consumption, APC: Amount of parent–child conversation time, CSS: Children’s somatic symptoms; *** *p* < 0.001, * *p* < 0.05.

**Table 5 ijerph-19-12975-t005:** Verification of the research model.

**Final Model**	** *x* ^2^ **	** *df* **	**TLI**	**CFI**	**RMSEA**
	108.6356	20	0.9395	0.9731	0.0412

**Table 6 ijerph-19-12975-t006:** Analysis of the path coefficients of the research model.

Pathway of Variables			Standardized Coefficients (β)	Unstandardized Coefficients (B)	Standard Error	C.R	*p*
CFBC (initial)	→	APC (initial)	0.1723	0.1168	0.0239	4.8798	***
CFBC (initial)	→	APC (rate of change)	0.0820	0.0255	0.0156	1.6331	0.1024
CFBC (rate of change)	→	APC (rate of change)	0.3227	0.2114	0.0380	505611	***
CFBC (initial)	→	CSS (initial)	−0.2332	−0.0592	0.0091	−6.5065	***
CFBC (initial)	→	CSS (rate of change)	−0.0003	0.0000	0.0058	−0.0061	0.9951
CFBC (rate of change)	→	CSS (rate of change)	−0.2174	−0.0514	0.0178	−2.8942	**
APC (initial)	→	CSS (initial)	−0.2524	−0.0945	0.0147	−6.4112	***
APC (initial)	→	CSS (rate of change)	−0.0671	−0.0111	0.0101	−1.1014	0.2707
APC (rate of change)	→	CSS (rate of change)	−0.6262	−0.2262	0.0495	−4.5684	***

CFBC: Children’s frequency of breakfast consumption, APC: Amount of parent–child conversation time, CSS: Children’s somatic symptoms; ** *p* < 0.01, *** *p* < 0.001.

**Table 7 ijerph-19-12975-t007:** Longitudinal mediating effect of the multivariate latent growth model (Sobel test).

Parameter Value	Z
children’s frequency of breakfast consumption (initial) → parent–child conversation time (initial) → children’s somatic symptom (initial)	−6.5913 ***
children’s frequency of breakfast consumption (rate of change) → parent–child conversation time (rate of change) → children’s somatic symptom (rate of change)	−12.5192 ***

*** *p* < 0.001.

## Data Availability

The following are available online at https://www.nypi.re.kr/archive/mps/program/examinDataCode/dataDwloadAgreeView?menuId=MENU00226 (accessed on 28 March 2022).

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
