# Peer review of "The Relationship between the Frequency of Breakfast Consumption, Conversation with Parents, and Somatic Symptoms in Children: A Three-Wave Latent Growth Model"

_ijerph, 2022, doi:10.3390/ijerph191912975_

Round 1
Reviewer 1 Report
Introduction:
This part is not clear:
Compared to adults, it is difficult for children and adolescents to express their emotions through language, making them vulnerable to emotional problems [13]. Parents and children share one of the most complex biological, psychological, and social relationships, and parents greatly influence children’s emotionality, especially as they help their children learn emotion regulation [14].
Please revise. For the phase “making them vulnerable to emotional problems”, should it be physical problems instead? I thought this part was referring to somatic symptoms.
In fact, your introduction touched on emotional problems to quite a large extent, but your paper should have focused on somatic symptoms. For example, this sentence “Since parents have the ability to exert a positive influence on children's emotional problems, it is important to secure frequent parent-child interactions in daily life.”, my next question would be “how about children’s somatic symptoms”; your argument can be made more direct so that readers can follow more easily.
Method:
Please provide more details on data collection, e.g., who completed the questionnaires and how the data were collected.
The survey were 2,607 children in the 1st year (1,313 boys, 1,294 girls), 2,437 children in the 2nd year (1,221 boys, 1,216 girls), and 2,411 children (1,211 boys, 1,200 girls) in the 3rd year. <- please provide attrition rate if this is a repeated measure study. If they are separate cohort, please provide more information about how to process and combine the data before latent growth modeling procedure.
The change in CFBC/APC over the years was very small, why would you select latent growth modeling as the analysis method? The results could be easier to understand if alternative models were used.
Results:
Please add a demographic table showing the basic information of study participants such as their average age, gender, frequency of breakfast consumption, and amount of parent-child conversation time. This can better justify whether the data distribution is appropriate for the proposed analysis strategies.
For the longitudinal mediating effect results, why would you analyze the initial value of one variable as the predictor of the initial value of the outcome variable through the initial value of the proposed mediator? This would make it like a cross-sectional study. The same question is for the results on rate of change. Authors may wish to read more about mediation analysis to get a better understanding of the implications of mediation results.
Discussion:
Again, the focus of result interpretation seemed to be on emotional problems, for example, this sentence: Furthermore, eating breakfast has been found to have positive effects in terms of mental health [36], since eating breakfast together can help create love, warmth, and empathy in families [37].
But since the study outcome is somatic symptoms, the discussion should explain why communication with parents could lead to a reduction in somatic symptoms, rather than emotional problems or mental health in general. Otherwise, readers would wonder why the study measured and analyzed somatic symptoms but not psychological/emotional problems.
Author Response
Thank you for your comments
please check my file

Reviewer 2 Report
The authors of this interesting manuscript analyze the relationship between breakfast consumption, "conversation with parents" and somatic symptoms in Korean children.
The manuscript is well organized and the authors provide adequate background for the research questions addressed. However, there are a few issues the authors will need to clarify.
Introduction
.- On page 1, lines 40-43, the authors report that the proportion of parents who talk to their children everyday is well below the average in OECD countries. As it seems that this analysis can be culturally sensitive or country specific, it would be more adequate to modify the title accordingly to metion the study was conducted on Korean children.
.- In the introduction, page 2, line 47, the authors report that 47,5% of the conversation time between children and parents occur during meals. It would be adequate if the authors could add some additional background information about frequency of family meals (parents and children) in Korea and what meal time is most frequently shared (breakfast? dinner? weekend meals?)
Minor: On page 1, lines 30-32, probably the authors mean "Breakfast consumption has been a major component of children's diet ...,"
Methods
.- They should clearly specify in the methods section the study design. In the present version of the manuscript it is not clear wether they analyzed data from three cross-sectional studies or they conducted a longitudinal follow-up of the same sample over three years. They mention that was the case on lines 327-328 and lines 334-335 at the end of the disccusion and conclusions, and also in the abstract, but it is not clear on the methods section.
Parent-child conversation (pages 3-4, lines 113-117), please specify which is the range of the computed score.
Results
CFBC: Children’s frequency of breakfast consumption: please for clarity, specify it is presented as number of days per week.
Please, delete lines 229-231 on page 7.
The title of figure 2 should be more specific to help understand the figure without reading the text. Add suitable footnotes to identify the information presented.
Most relevant information presented on table 5 is already presented on figure 2. I would suggest to move information on this table to supplementary material; modify the title and add footnotes accordingly to figure 2.
Author Response

(The authors gave the same response as above.)

Round 2
Reviewer 1 Report
Nil